# Identification of Peptides from Edible *Pleurotus eryngii* Mushroom Feet and the Effect of Delaying D-Galactose-Induced Senescence of PC12 Cells Through TLR4/NF-κB/MAPK Signaling Pathways

**DOI:** 10.3390/foods13223668

**Published:** 2024-11-18

**Authors:** Fen Zhao, Ji’an Gao, Haiyan Li, Shuaishuai Huang, Shangmeng Wang, Xinqi Liu

**Affiliations:** Key Laboratory of Geriatric Nutrition and Health Ministry of Education, Beijing Technology and Business University, Beijing 100048, China

**Keywords:** *Pleurotus eryngii*, edible mushroom feet peptides, simulated digestion in vitro, senescence, PC12 cells, TLR4/MAPK/NF-κB signaling pathways

## Abstract

*Pleurotus eryngii* mushroom has been proven to have anti-aging bioactivities. However, few studies have focused on edible *Pleurotus eryngii* mushroom feet peptides (PEMFPeps). In this paper, the effects of delaying the senescence of D-Galactose-induced PC12 cells were evaluated, and the mechanisms were also investigated. PEMFPeps were prepared by alkaline protease enzymolysis of edible *Pleurotus eryngii* mushroom feet protein (PEMFP), which mainly consisted of a molecular weight of less than 1000 Da peptides, primarily occupying 89.15% of the total. Simulated digestion in vitro of *Pleurotus eryngii* mushroom feet peptides (SID-PEMFPeps) was obtained in order to further evaluate the bioactivity after digestion. The peptide sequences of PEMFPeps and SID-PEMFPeps were detected by LC-MS/MS subsequently. Five new peptides of PEMFPeps and one new peptide of SID-PEMFPeps were identified. The effects of PEMFP, PEMFPeps, and SID-PEMFPeps on D-Galactose-induced senescence of PC12 cells were evaluated. PEMFP, PEMFPeps, and SID-PEMFPeps could all enhance antioxidant enzyme activities significantly, such as superoxide dismutase (SOD), glutathione peroxidase (GSH-Px), and catalase (CAT); decrease the intracellular levels of malondialdehyde (MDA) and reactive oxygen species (ROS); and inhibit the senescence-associated β-galactosidase (SA-β-gal) activity, among which SID-PEMFPeps showed the best effects. Western blotting analysis confirmed that SID-PEMFPeps significantly regulated the expressions of key proteins such as TLR4, IKKα, IκBα, p65, ERK, and JNK1/2/3, which indicated that SID-PEMFPeps could delay D-Gal-induced senescence of PC12 cells through TLR4/NF-κB/MAPK signaling pathways. This is the first time to investigate PEMFPeps and SID-PEMFPeps protective effects and mechanisms. Our study could lay a solid foundation for PEMFPeps to be used as nutritional supplementation to reduce aging-related damage. And the application of PEMFPeps could also provide optional solutions in exploring more edible protein resources for human beings.

## 1. Introduction

Aging is an irreversible process that occurs at the cellular, organ, and whole-organism levels, leading to disruptions in physiological functions and the onset of related degenerative diseases within the organism [1]. The mutations in mitochondrial DNA (mtDNA) alter expressions of oxidative phosphorylation complexes (OxPhos) and result in mitochondrial dysfunctions, leading to the rapid generation of reactive oxygen species (ROS) [2]. As time passes, biological functions become weaker, and the capacities of organisms to mitigate oxidative stress diminish. Accumulation of excessive ROS could result in impairment of cellular structures and metabolic processes [3]. The damage usually leads to oxidative stress and inflammation, subsequently causing cellular aging and death, which expedites the aging process [4]. In view of the economic and psychological burdens endured by individuals with age-related degenerative diseases, pursuing methods to prevent or mitigate the impact of aging has gained global attention [5].

Edible fungi are rich in bioactive metabolites, including protein, polysaccharides, enzymes (such as superoxide dismutase), dietary fiber, and other components. And they could be excellent raw materials for functional foods and nutritional supplements to neutralize free radicals [6]. Compared to anti-aging drugs, peptides derived from natural edible fungi have significant advantages in anti-aging, such as high activity, low molecular weight, and great bioavailability [7,8]. Edible fungi peptides have become popular as emerging ingredients of dietary supplements and pharmaceuticals in recent years for modulating inflammatory and apoptosis pathways, activating immune cells, and improving age-related degenerative conditions such as neurodegenerative disorders, inflammation, hyperglycemia, hypertension, and hypercholesterolemia [9,10]. For instance, *Cordyceps militaris* peptides improved memory and reduced cognitive decline in mice. AChE is a neurotransmitter that is strongly related to learning and memory functions. The increased AChE concentration could damage the cholinergic nerves and lead to neurological diseases like cognitive impairments. *Cordyceps militaris* peptides could treat neurodegenerative diseases by clearing ROS and reducing AChE activity in the mouse brain. They acted as both free radical scavengers and AChE inhibitors in neural cells [11,12]. *Tricholoma matsutake*-derived peptides WFNNAGP regulated the expression of NF-κB, inhibited NLRP3 and caspase-1 formation and activation, and restrained pro-inflammatory factors to alleviate murine colitis [13].

*Pleurotus eryngii*, commonly known as king oyster mushroom, could be used as both medicine and food, owing to its remarkable nutritional and therapeutic components, such as protein, free amino acids, and phytochemicals. Pharmacological investigations demonstrated that peptides derived from *Pleurotus eryngii* possessed various biological activities, including antioxidant, anti-inflammatory, anti-aging, anti-tumor, and immunomodulatory, which have application potential in functional foods. Research on *Pleurotus eryngii* mycelium peptides revealed that PEMP scavenged free radicals, inhibited cancer cell proliferation, promoted macrophage proliferation and phagocytosis, and enhanced cytokine secretion. It was a good natural anti-aging source and antioxidant with anti-tumor and immunostimulatory activities [14]. *Pleurotus eryngii* peptides have been shown to scavenge excess free radicals and promote antibody, cytokine, and chemokine production, enhancing the oxidative defense system and barrier function, which contribute to extending lifespan and reducing the presence of aging-related markers in the organisms [15]. *Pleurotus eryngii* mushroom feet are edible by-products, the protein nutritional value of which is similar to that of commercial mushrooms. However, most of them were thrown away in the edible fungi industry, which was a waste of protein resources.

Thus, in our study, *Pleurotus eryngii* mushroom feet protein (PEMFP), *Pleurotus eryngii* mushroom feet peptides (PEMFPeps), and simulated digestion in vitro of *Pleurotus eryngii* mushroom feet peptides (SID-PEMFPeps) were prepared. The effects of delaying senescence of D-Galactose-induced PC12 cells were evaluated, and the mechanisms of SID-PEMFPeps were also investigated. Our study will provide solid theoretical foundations for utilizing PEMFPeps in functional foods to alleviate aging and age-related degenerative diseases.

## 2. Materials and Methods

### 2.1. Materials and Chemicals

*Pleurotus eryngii* mushroom feet were obtained from Xuzhou, China. Alkaline protease, D-Galactose (D-Gal), and TRIzol^®^ reagents were procured from Shanghai YuanYe Biological Technology Co., Ltd. (Shanghai, China). Dialysis bags of molecular weight cutoff (MWCO) 200 Da were acquired from Microdyn-Nadir GmbH (Microdyn-Nadir GmbH, Wiesbaden, Germany). PC-12 cells (4th generation) were procured from Shanghai Fuheng Biological Technology Co., Ltd. (Shanghai, China). RPMI 1640 medium (1640), penicillin–streptomycin (PS), phosphate-buffered saline (PBS), and trypsin-EDTA (TE) were procured from Gibco (Grand Island, NE, USA). Fetal bovine serum (FBS) was obtained from Excell Biotechnology Co. (Taicang, China). Horse serum (HS), RIPA lysis buffer, and BCA protein concentration determination kit were procured from Beijing Solarbio Science & Technology Co., Ltd. (Beijing, China). ROS detection assay kit, SA-β-Gal staining kit, and Cell Counting Kit-8 (CCK-8) kit were sourced from Beyotime Biotechnology Co., Ltd. (Shanghai, China). Glutathione peroxidase (GSH-Px) and superoxide dismutase (SOD) detection assay kits were procured from Suzhou Grace Biotechnology Co., Ltd. (Suzhou, China). Catalase (CAT) and malondialdehyde (MDA) detection assay kits were acquired from Nanjing Jiancheng Bioengineering Institute (Nanjing, China). In addition, antibodies for TLR4, IKK, p65, IκB, ERK, JNK1/2/3, GAPDH, and α-Tubulin; horseradish peroxidase-conjugated goat anti-rabbit IgG (H+L); Western secondary antibody dilution; and BeyoECL Star (extra super-sensitive enhanced chemiluminescence kit) were procured from Beyotime Biotechnology (Shanghai, China).

### 2.2. Preparation of PEMFP, PEMFPeps, and SID-PEMFPeps

The preparation method of PEMFP, PEMFPeps, and SID-PEMFPeps are shown in Figure 1. The fresh *Pleurotus eryngii* mushroom feet were washed three times and stored at −40 °C. Freeze-dry them for 48 h by using a vacuum freeze dryer. The dried mushroom feet were ground into powder using a food grinder and sieved for storage. After that, PEMFP was extracted using the salt dissolution method with ultrasonic assistance. Alkaline protease was employed for enzymatic hydrolysis to obtain PEMFPeps. Simulated digestion in vitro was carried out to prepare SID-PEMFPeps.

#### 2.2.1. Preparation of PEMFP

Approximately 10 mg of freeze-dried *Pleurotus eryngii* mushroom feet powder was suspended in 4% NaCl solution at 1:45. Then, extraction at 30 °C for 30 min was assisted by ultrasonic. The *Pleurotus eryngii* mushroom feet protein solution was obtained by centrifugation for 15 min at 8000 rpm. Subsequently, it was dialyzed against a 200 Da MWCO dialysis bag for 24 h to remove salts. The solution was stored at −40 °C and subjected to freeze-drying for 48 h by the vacuum freeze dryer to obtain PEMFP powder.

#### 2.2.2. Preparation of PEMFPeps

The lyophilized PEMFP powder was reconstituted in boiling water and adjusted to pH 9.0 by NaOH. The alkaline protease concentration was adjusted to 0.96%, and enzymatic hydrolysis was conducted at 66 °C for 2.9 h. The enzyme reaction was halted in a water bath for 15 min at 95 °C. The supernatant was collected by centrifugation at 4500 rpm for 20 min, followed by dialysis for 24 h with a 200 Da MWCO dialysis bag. Subsequently, vacuum freeze-drying for 48 h yielded PEMFPeps.

#### 2.2.3. Preparation of SID-PEMFPeps

The method of preparing SID-PEMFPeps was referenced from the study of Brodkorb with modifications [16]. The simulated digestion in vitro process began with oral digestion, where PEMFPeps were added to 10 mL of simulated saliva fluid and placed on a shaking incubator at 37 °C for 2 min. The oral-digested samples were mixed with 20 mL of simulated gastric fluid, and the pH was adjusted to 3.0. Pepsin was added to achieve 2000 U/mL in the mixture, and digestion was carried out at 37 °C for 2 h. This was followed by intestinal digestion, where the gastric-digested samples were mixed with 40 mL of simulated intestinal fluid, and the pH was adjusted to 7.0. Pancreatic enzymes were added to achieve the activity level of 100 U/mL in the mixed solution, and digestion was conducted at 37 °C for 2 h. Finally, the digestive fluids were soaked in boiling water for 10 min to halt digestion. Afterward, they were centrifuged at 4500 rpm for 30 min at 4 °C. Once centrifugation was complete, the supernatant was collected and stored at −40 °C. The SID-PEMFPeps were obtained by freeze-drying the supernatant in a vacuum freeze dryer for 48 h.

### 2.3. Identification Methods of Peptide Sequences of PEMFPeps and SID-PEMFPeps

#### 2.3.1. Sample Preparation

After reduction with 10 mM dithiothreitol (DTT) for 1 h at 56 °C, then alkylation in the dark at room temperature for 40 min using 50 mM iodoacetamide (IAA). The extracted peptides were freeze-dried until dry. The samples were reconstituted with 20 μL 0.1% formic acid prior to LC-MS/MS analysis.

#### 2.3.2. Nano LC-MS/MS Analysis

The chromatographic column was 150 μm × 15 cm and packed with C18 material (1.9 μm, 100 Å, Dr. Maisch GmbH, Germany) at a flow rate of 600 nL/min. A mobile phase consisting of solvent A (0.1% formic acid aqueous solution) and solvent B (20% 0.1% formic acid aqueous solution—80% acetonitrile) was chosen for separating the peptides. The chromatographic gradient conditions were as follows: 4% to 8% B over 3 min, 8% to 28% B over 86 min, 28% to 40% B over 20 min, 40% to 95% B over 1 min, and 95% B held for 10 min. The injection volume was 5 μL. The parameters for MS analysis were set as follows: resolution 70,000, AGC target 3 × 10^6^, maximum IT 100 ms, and scan range from 100 *m*/*z* to 1500 *m*/*z*. For MS/MS analysis, the parameters were as follows: resolution 17,500, AGC target 1 × 10^5^, maximum IT 50 ms, TOPN 20, and NCE/Step NCE 28. Evaluation of raw MS files and search of protein databases by sample type were carried out using the Byonic 4.2.4 software.

### 2.4. Cultivation of PC12 Cells and Construction of the Senescent Cell Model

#### 2.4.1. Cultivation of PC12 Cells

The rat gland pheochromocytoma (PC12) cells were obtained from Fuheng Life Sciences (Shanghai, China), and the cell information was recorded in Cell Bank, Chinese Academy of Science, serial number SCSP-517. The cells were cultured in a medium of 85% RPMI 1640, 5% fetal bovine serum (FBS), 10% horse serum (HS), and 1% penicillin–streptomycin [17]. Cultivation was performed in the CO_2_ incubator at 37 °C with a 5% CO_2_ concentration for 24 h. The cells were cultured to reach 80–90% capacity and then inoculated into 96-well plates and 6-well plates for experiments.

#### 2.4.2. Cell Viability Assay and Construction of Senescent Cell Model Induced by D-Gal

First, PC12 cells were exposed to different concentrations of D-Gal to determine the appropriate D-Gal concentration for senescence induction. Logarithmic growth phase cells were taken, digested by tryptase (TE), made into cell suspension, and inoculated into 96-well plates at 1 × 10^5^ cells/mL incubated overnight at 37 °C in 5% CO_2_ incubator. Then, D-Gal was added at different concentrations (2.5, 5, 10, 15, 20, and 25 mg/mL) as well as in the control group without D-Gal, and the cells were incubated for 24 h to evaluate the cytotoxicity of D-Gal on PC12 cells.

Next, the appropriate sample concentration was determined to avoid cell damage caused by GSH, PEMFP, PEMFPeps, and SID-PEMFPeps. Logarithmic growth phase cells were taken, digested by tryptase (TE), made into cell suspension, and inoculated into 96-well plates at 1 × 10^5^ cells/mL incubated overnight at 37 °C in 5% CO_2_ incubator then added D-Gal and incubated for 24 h. Different concentrations (0.25, 0.5, 1, and 2.5 mg/mL) of GSH, PEMFP, PEMFPeps, and SID-PEMFPeps were added; control group did not have D-Gal or samples; model group with D-Gal had no samples and was incubated for 24 h to evaluate the cytotoxicity of the samples on PC12 cells. Cell viability was assessed using the CCK-8 assay kit from Beyotime Biotechnology Co., Ltd., Shanghai, China.

### 2.5. Determination of ROS

PC12 cells from different groups were treated according to the instructions received from the manufacturer Beyotime (Shanghai, China). The culture medium from the 6-well plate was removed, and the DCFH-DA probe was added with a 10 μmol/L final concentration after 1:1000 dilution. Subsequently, the plates were incubated in the dark at 37 °C for 30 min. The cells were washed three times with PBS to remove any DCFH-DA that did not enter the cells completely. Afterward, the cells were digested for 2 min using TE, followed by aspiration and mixing before transferring to a black opaque 96-well plate for assay. For observing and photographing the cells, a fluorescence-inverted microscope was used at 485 nm excitation wavelength and 535 nm emission wavelength.

### 2.6. Protective Effects of PEMFP, PEMFPeps, and SID-PEMFPeps on Senescent PC12 Cells

The cells were digested with TE during the logarithmic growth phase, making cell suspension, and inoculated into 6-well plates homogeneously. PC12 cells were adhered overnight and treated with different mediums for 24 h. The blank and model groups were treated with a medium without GSH and samples, the positive control group was treated with a medium containing GSH, and the sample groups were treated with a medium containing different concentrations of PEMFP, PEMFPeps, and SID-PEMFPeps. Subsequently, cells were senescence damaged with 1640 containing D-Gal without FBS and HS for 24 h. Cells were rinsed with PBS, and 100 μL of cell lysis buffer containing 1% PMSF was added into wells. The lysis solution was blown several times, which ensured complete contact with cells. Then, place the 6-well plate on ice for 30 min. SOD and GSH-Px activities were determined according to the instructions of the Suzhou Geruisi assay kit (Suzhou, China). CAT and MDA levels were measured following the instructions of the Nanjing Jiancheng assay kit (Nanjing, China).

### 2.7. Senescence-Associated β-Galactosidase (SA-β-Gal) Staining

SA-β-gal staining experiments were performed with the β-galactosidase staining kit (Beyotime, Shanghai, China), following the instructions of the manufacturer. Staining images were taken under the microscope (IX 73, OLYMPUS, Tokyo, Japan), and the ratio of SA-β-gal senescence-positive cells was calculated.

### 2.8. Western Blotting Analysis

The Western blot assay was referenced and modified from the work of Chen et al. [18]. The culture medium was removed from 6-well plates, cells were rinsed with PBS, and then different groups of PC12 cells were treated using 100 µL lysis buffer with 1% PMSF and phosphatase inhibitor. The lysis buffer was pipetted several times to ensure thorough contact with cells. Then, 6-well plates were frozen for 30 min, with intermittent shaking every 5 min. After complete lysis, the samples were centrifuged at 10,000× *g* for 5 min to collect the supernatant. Protein concentration and dilution were determined using the BCA assay to standardize. The samples were mixed with 5× loading buffer, heated at 100 °C for 5 min, and centrifuged at 10,000× *g* for 2 min to collect the supernatant for analysis. Protein separation was performed using 10% SDS-PAGE, followed by transfer to a PVDF membrane. They were then blocked for 1 h at room temperature with 5% skim milk, and then the membrane was washed three times with TBST. Primary antibodies (TLR4, IKK, P65, IκBα, JNK1/2/3, ERK, GAPDH, α-Tub, 1:1000) were incubated overnight at 4 °C, followed by three washes with TBST. Secondary antibodies (anti-rabbit IgG (H+L), 1:1000) were incubated at room temperature for 1 h. GAPDH and α-Tub were used as loading controls. Band intensity was observed by ECL chemiluminescence and measured by Image J v1.8.0 software.

### 2.9. Data Analysis

Data were statistically analyzed and expressed as mean ± standard deviation (SD). The results were analyzed using a single-factor analysis of variance (ANOVA) and Tukey’s post hoc test using SPSS 24 statistical software. The *p*-value less than 0.05 was considered statistically significant.

## 3. Results

### 3.1. Protein Content, Hydrolysis Degree, and Molecular Weight Distribution of PEMFPeps

According to the literature and preliminary experiments, solid–liquid ratio, extraction time, extraction temperature, and NaCl solution concentration were considered the main factors that influenced the extraction rate of PEMFP. And pH, alkaline protease concentration, enzymatic hydrolysis temperature, and enzymatic hydrolysis time were considered the main factors that influenced the extraction rate of PEMFPeps. Single-factor and response surface experiments were conducted to obtain the optimal conditions.

Subsequently, the protein content, hydrolysis degree, and molecular weight distribution of PEMFPeps were determined. The protein content of PEMFPeps was 46.246 ± 1.864/100 g, and the hydrolysis degree finally reached 68%. As shown in Figure 2, the molecular weights of less than 1000 Da were approximately 89.15%, and those of more than 1000 Da were about 10.85%. As reported, protein hydrolysates were typically a mixture of low-molecular-weight peptides, and molecular weight has a significant influence on their anti-aging and anti-inflammatory activities [19,20]. Cao et al. obtained anti-skin aging peptides less than 3 kDa by enzymatic hydrolysis of chicken bone collagen, which accounted for about 87.61% of the total hydrolysate [21]. Qiu et al. obtained the anti-aging peptides mixture, where 95.8% had molecular weights less than 1 kDa [22]. Therefore, through the result that PEMFPeps are a mixture of small-molecule peptides, it is speculated that PEMFPeps probably have antioxidant activity.

### 3.2. Construction of Senescent Cell Model Induced by D-Gal

PC12 cells were incubated with different concentrations of D-Gal (2.5, 5, 10, 15, 20, and 25 mg/mL) for 24 h, and the cell viability was measured to ascertain the concentration of D-Gal-induced senescence. The results are shown in Figure 3. As the concentration of D-Gal increased, PC12 cell viability decreased in a concentration-dependent manner, and cell viability was less than 60% at a concentration exceeding 20 mg/mL. This indicated that cytotoxicity was excessive at more than 20 mg/mL D-Gal concentration, and the cells suffered excessive damage and death, which might adversely affect the following experiments. Therefore, 20 mg/mL was selected as the optimal concentration for establishing a senescent PC12 cell model induced by D-Gal.

### 3.3. Protective Effects on Cell Viability

Before investigating the preventive effects of PEMFP, PEMFPeps, and SID-PEMFPeps on D-Gal-induced senescence in PC12 cells, it was necessary to determine their cytotoxicity to avoid cellular damage due to the addition of samples. Cells were treated with different concentrations (0.25, 0.5, 1, and 2.5 mg/mL) of PEMFP, PEMFPeps, SID-PEMFPeps, and GSH for 24 h, respectively. The results are shown in Figure 4. The PC12 cell viabilities of PEMFP, PEMFPeps, and SID-PEMFPeps were not inhibited at 0.25, 0.5, and 1 mg/mL compared to the control. However, cell viability was inhibited at the concentration of 2.5 mg/mL. Beyond that, all concentrations of GSH did not inhibit cell viability. So, the appropriate concentration should be less than 2.5 mg/mL for subsequent experiments to conduct nutritional interventions in order to prevent damage to cells resulting in experimental inaccuracies. Considering the experimental results, experimental reproducibility, and the difficulty of experimental operation, 0.25, 0.5, and 1.0 mg/mL were selected as the low, medium, and high concentrations of PEMFP, PEMFPeps, and SID-PEMFPeps. And 1.0 mg/mL was selected as the GSH-positive control concentration.

### 3.4. Expression of ROS Level

Low concentrations of ROS are beneficial for normal cell growth, but an excess of ROS can damage cell structures, leading to oxidative stress [23]. The 2′,7′-Dichlorodihydrofluorescein diacetate (DCFH-DA) can pass through the cell membrane freely and hydrolyze to 2′,7′-dichlorodihydrofluorescein (DCFH), accumulating in cells. ROS can oxidize DCFH to 2′,7′-dichlorofluorescein (DCF) and emit green fluorescence. Therefore, the ROS levels of different concentrations of PEMFP, PEMFPeps, and SID-PEMFPeps groups were determined by measuring ROS fluorescence intensity. As shown in Figure 5, ROS levels in the model group increased rapidly after treatment with D-Gal (20 mg/mL) for 24 h, with a 4.55-fold increase in fluorescence intensity compared to the control group (*p* < 0.05). In PEMFP pretreated groups, the PEMFP-L and PEMFP-M groups decreased ROS fluorescence intensity by 31.41% and 12.45%, respectively, while the PEMFP-H group elevated by 7.46%. The greater effectiveness in reducing ROS levels in the PEMFP-L group may be due to the fact that PEMFP is a protein with higher molecular weight, and the cells have limited ability to absorb the protein. However, the higher concentration of PEMFP inhibited the effect of reducing the ROS level. Pretreatment with PEMFPeps and SID-PEMFPeps (0.25, 0.5, and 1.0 mg/mL) significantly suppressed ROS generation, of which PEMFPeps-H and SID-PEMFPeps-H showed the best effects and reduced by 36.26% and 51.53%, respectively (*p* < 0.05). These data demonstrated that PEMFP, PEMFPeps, and SID-PEMFPeps were able to eliminate the accumulation of intracellular ROS, with the SID-PEMFPeps-H group showing the greatest scavenging activity, which could effectively inhibit oxidative stress and alleviate D-Gal-induced PC12 cell senescence.

### 3.5. Protective Effects on the Antioxidant System of Senescent PC12 Cells

The levels of SOD, GSH-Px, CAT, and MDA in PC12 cells were determined to estimate the antioxidant activities of PEMFP, PEMFPeps, and SID-PEMFPeps on D-Gal-induced oxidative damage. As shown in Figure 6, compared with the control group, the levels of SOD, GSH-Px, and CAT in the model group decreased by 58.99%, 63.58%, and 52.23%, respectively, while the MDA content increased to 149.17% (*p* < 0.05). This indicated that oxidative stress was enhanced in D-Gal-induced PC12 cells. Compared with the model group, all of the PEMFP, PEMFPeps, and SID-PEMFPeps groups increased SOD, GSH-Px, and CAT activities and decreased MDA content. The SOD content of PEMFP-L, PEMFPeps-H, and SID-PEMFPeps-H groups increased by 31.99%, 43.73%, and 53.65% (*p* < 0.05), respectively (Figure 6A). GSH-Px levels of PEMFP-H, PEMFPeps-H, and SID-PEMFPeps-H groups were enhanced by 49.47%, 64.92%, and 77.38% (*p* < 0.05), respectively (Figure 6B). The CAT activity of PEMFP-M, PEMFPeps-H, and SID-PEMFPeps-H groups increased by 34.50%, 33.16%, and 36.51% (*p* < 0.05) (Figure 6C). And the MDA content of PEMFP-H, PEMFPeps-H, and SID-PEMFPeps-H groups decreased by 21.74%, 25.4%, and 27.11% (*p* < 0.05) correspondingly (Figure 6D).

The results showed that the SOD, GSH-Px, and CAT levels were significantly increased, and MDA content was significantly decreased in senescent PC12 cells. Overall, the SID-PEMFPeps-H group exhibited the best antioxidative capacity. This may be attributed to factors such as molecular weight, amino acid composition, and sequence, as well as the hydrophobicity of specific amino acid residues of the SID-PEMFPeps. On the basis of the literature, lower molecular weight, and the existence of hydrophobic and aromatic amino acid residues, hydrophobic and hydrogen donor effects could be the reasons for the stronger antioxidant capacity of the peptides.

### 3.6. SA-β-gal Staining and the Rate of SA-β-gal-Positive Cells

SA-β-gal is an essential marker to indicate cellular senescence, and the SA-β-gal-positive cell number could indicate cellular senescence level. Cellular senescence produced numerous ROS, and mushroom peptides could scavenge excessive ROS and delay oxidative stress and senescence. As a result, the senescence-delaying effect of mushroom peptides can be indicated by SA-β-gal staining. As shown in Figure 7A, compared with the control group, PC12 cells in the model group were significantly enlarged and flattened, indicating a successful establishment of the senescent cells model. The rate of SA-β-gal-positive cells was significantly increased by 62.47% (Figure 7B). However, the rate of SA-β-gal-positive cells was markedly decreased with the treatment of different concentrations of PEMFP, PEMFPeps, and SID-PEMFPeps (0.25, 0.5, and 1.0 mg/mL) compared to the model group. All of the PEMFP, PEMFPeps, and SID-PEMFPeps groups reduced oxidative stress-induced accumulation of senescent cells significantly. And there was no significant difference within the groups. The positive staining rates of PEMFP-H, PEMFPeps-H, and SID-PEMFPeps-H were significantly decreased by 33.27%, 38.72%, and 39.86%, respectively. Among these groups, the SID-PEMFPeps-H group demonstrated the most efficient activity. These results demonstrated that PEMFP, PEMFPeps, and SID-PEMFPeps showed the effects of relieving cellular senescence.

### 3.7. Identification Results of Peptide Sequences of PEMFPeps and SID-PEMFPeps

The main peptides of PEMFPeps and SID-PEMFPeps were identified, respectively, and scores were used to represent the accuracy. The MS spectra of PEMFPeps and SID-PEMFPeps are shown in Figure 8A,B. Peptide sequences are shown in Table 1 and Table 2. The seven main peptides in the PEMFPeps were selected, in which five new peptides were identified, including GHGFEGVTH, GRHTGPGKR, DDESAIGIR, HTGNIPLDE, and DACLPSPK. The seven main peptides in the SID-PEMFPeps were selected, in which one new peptide, HVPL, was identified. The MS spectra of main peptides GHGFEGVTH of PEMFPeps and VPIIPL of SID-PEMFPeps are shown in Figure 8C,D. The composition of amino acids is intimately related to antioxidant activity, in which hydrophobic, aromatic, and polar amino acids contribute to the anti-aging activities of peptides [24]. Hydrophobic amino acids can enhance peptide solubility in the lipid phase through increased interaction between peptides and polyunsaturated fatty chains in biological membranes. This effect promoted peptide interactions with free radicals, leading to an enhanced ability of peptides to suppress lipid peroxidation. In addition, the presence of the N-terminus of hydrophobic amino acids promoted the interactions of hydrogen donors and other amino acids, thereby enhancing its anti-aging potency [24]. Polar amino acids are hydrogen donors with strong free radical scavenging activity [25,26], which can block free radical oxidation by metal ions with charged side-chain groups as a complex, thus chelating the metal ions to exhibit good anti-aging activity [27]. Due to the structures of aromatic amino acids and the presence of phenolic groups, the residues can easily release protons into electron-deficient free radicals, effectively eliminating free radicals and ultimately producing powerful anti-aging activity [28,29]. Therefore, hydrophobic, polar, and aromatic amino acids served as key amino acids in peptide screening. The anti-aging potentials of specific amino acids such as hydroxyproline (Hyp), leucine (Leu), alanine (Ala), and valine (Val) have been demonstrated, which established their feasibilities for application as anti-aging agents [24].

As shown in Table 1 and Table 2, peptide sequences mainly contain key amino acids (hydrophobic, aromatic, and polar). The peptides included hydroxyproline (Hyp), leucine (Leu), alanine (Ala), valine (Val), phenylalanine (Phe), threonine (Thr), and other hydrophobic, aromatic, and polar amino acids. Hydrophobic residues promoted peptide solubility at the lipid–water interface, aromatic residues supplied hydrogen protons, and polar residues influenced the metal ion chelating capacity of peptides. They facilitated free radical scavenging, and it could be speculated that these amino acids together contributed to the significant activity of delaying D-Gal-induced senescence in PC12 cells.

### 3.8. Effect of SID-PEMFPeps on TLR4/NF-κB/MAPK Signaling Pathways

Aging is primarily associated with cascades of oxidative stress, inflammation, and apoptosis-related signaling pathways. TLR4 is a primary membrane signaling receptor leading to inflammation, with signaling pathways such as nuclear factor-κB (NF-κB) and mitogen-activated protein kinases (MAPK) playing potential roles in it [30]. NF-κB acted as a transcription factor during the aging process and was associated with many age-related diseases [31], and its signaling was upregulated in conditions such as osteoarthritis, atherosclerosis, neurodegenerative diseases, cardiovascular diseases, and atherosclerosis [32]. The IκB family of proteins keeps NF-κB transcription factors in an inactive state in the cytoplasm. Phosphorylation of NF-κB by IKK leads to the activation and initiation of transcriptional responses [31]. Oxidative stress can elevate ROS levels and activate TLR4/NF-κB pathway, causing the release of NF-κB from IκBα and subsequently leading to its nuclear translocation [24,33]. As NF-κB signaling was activated, phosphorylation of p65 promoted nuclear translocation of p65 subunits, enhancing the transcriptional activity of inflammatory mediators [34]. NF-κB p65 signaling pathway was involved in the occurrence and progress of aging and its associated diseases, as well as playing an important role in it [35]. MAPK is a signaling pathway that participates in numerous physiological processes. The ERK cascade reaction was the first described MAPK pathway, and its phosphorylation activated MAPK signaling [20]. p-JNK pathway activation could lead to D-Gal-induced neuronal apoptosis. Oxidants and peroxides can also activate p-JNK by generating ROS, thereby accelerating cell apoptosis and promoting tissue aging [36].

Figure 9A illustrates the Western blotting analysis results of TLR4/NF-κB pathway-related proteins in the control, model, SID-PEMFPeps-L, SID-PEMFPeps-M, and SID-PEMFPeps-H groups. As shown in Figure 9B, the expression level of TLR4 had a dose-dependent manner with the concentration increased. Compared with the control group, the expression level of TLR4 was significantly increased by 28.84% in the model group (*p* < 0.05). And compared with the model group, the TLR4 protein expression levels of SID-PEMFPeps-L, SID-PEMFPeps-M, and SID-PEMFPeps-H groups were significantly inhibited by 19.17%, 33.73%, and 48.04% (*p* < 0.05), respectively. Surprisingly, the SID-PEMFPeps-M and SID-PEMFPeps-H groups had no significant difference or lower expression levels than the control group. The expression levels of IKKα in different groups are indicated in Figure 9C. In comparison with the control group, the IKKα expression level of the model group was significantly inhibited by 53.05%. Compared with the model group, the IKKα expression levels of SID-PEMFPeps-L, SID-PEMFPeps-M, and SID-PEMFPeps-H groups were significantly increased by 16.52%, 35.40%, and 40.03% (*p* < 0.05), respectively. And the SID-PEMFPeps-M and SID-PEMFPeps-H groups exhibited almost the same expression level as the control group. Figure 9D showed that the expression level of p65 was significantly increased by 37.28% in the model group compared to the control group. In comparison with the model group, the p65 protein expression levels of SID-PEMFPeps-L, SID-PEMFPeps-M, and SID-PEMFPeps-H groups were significantly inhibited by 8.71%, 20.02%, and 35.38%, respectively (*p* < 0.05). The results in Figure 9E revealed that the IκBα expression level of the model group was significantly inhibited by 29.27% compared with the control group (*p* < 0.05). Compared with the model group, the IκBα protein expression levels of SID-PEMFPeps-L, SID-PEMFPeps-M, and SID-PEMFPeps-H groups were significantly increased by 8.66%, 13.31%, and 21.56% (*p* < 0.05), respectively. Based on the above, the results indicated that SID-PEMFPeps could delay cellular senescence by affecting expression levels of key proteins in TLR4/NF-κB inflammatory pathways. The ROS was over-accumulated after D-Gal treatment, and the TLR4/NF-κB signaling pathways were greatly activated in PC12 cells. However, the nutritional intervention of SID-PEMFPeps reduced ROS accumulation, significantly inhibited the protein expression levels of TLR4 and p65, and increased the protein expression levels of IKKα and IκBα. The translocation process of p65 to the nucleus was delayed, which effectively suppressed IκBα to promote the release of NF-κB and inhibited its nuclear translocation.

Figure 10A illustrates the Western blotting analysis results of MAPK pathway-related proteins in the control, model, SID-PEMFPeps-L, SID-PEMFPeps-M, and SID-PEMFPeps-H groups. As shown in Figure 10B, compared to the control group, the ERK expression level was significantly inhibited by 31.56% in the model group. Compared with the model group, the ERK protein expression levels of SID-PEMFPeps-L, SID-PEMFPeps-M, and SID-PEMFPeps-H group were significantly increased by 11.12%, 25.90%, and 33.10%, respectively (*p* < 0.05). The ERK expression levels of SID-PEMFPeps-M and SID-PEMFPeps-H groups showed no significance with the control group. In Figure 10C, the expression level of JNK1/2/3 had a dose-dependent manner with the concentration increasing. The JNK1/2/3 protein expression level of the model group was significantly inhibited by 41.50% compared with the control group. In comparison with the model group, JNK1/2/3 protein expression levels of SID-PEMFPeps-L, SID-PEMFPeps-M, and SID-PEMFPeps-H groups were significantly increased by 13.14%, 35.40%, and 36.75%, respectively (*p* < 0.05). According to the above results, SID-PEMFPeps could affect the expression of key proteins in the MAPK signaling pathway and had the effect of delaying cellular senescence. D-Gal-induced ROS accumulation was effectively reduced after the nutritional intervention of SID-PEMFPeps, and the JNK1/2/3 and ERK phosphorylation were inhibited. Thus, the inflammatory response was attenuated, apoptosis was delayed, and cellular senescence was retarded.

To sum up, SID-PEMFPeps could delay cellular senescence by affecting expression levels of key proteins in TLR4/NF-κB/MAPK inflammatory pathways. Among different concentration groups, the SID-PEMFPeps-H group had the best effect.

## 4. Conclusions

In conclusion, about 89.15% of the PEMFPeps prepared in this study consisted of peptides with molecular weights less than 1000 Da, and the degree of hydrolysis eventually reached 68%. PEMFPeps may have a variety of biological activities. The main peptides of PEMFPeps and SID-PEMFPeps were identified by LC-MS/MS, while six new peptides were found. Peptide sequences indicated that PEMFPeps and SID-PEMFPeps were rich in hydrophobic, aromatic, and polar amino acids, which were related to anti-aging activities. PEMFP, PEMFPeps, and SID-PEMFPeps all exerted mitigating effects on the D-Gal-induced senescence of PC12 cells. They could efficiently eliminate the accumulation of ROS; increase the activities of antioxidant enzymes such as CAT, SOD, and GSH-Px; decrease the level of MDA; and reduce the rate of SA-β-Gal-induced positive senescent cells. Overall, the effect of SID-PEMFPeps was better than PEMFP and PEMFPeps, and the SID-PEMFPeps-H group had the best effect. After simulated digestion, the senescence-delaying activity of SID-PEMFPeps was obviously increased compared to PEMFPeps. Further Western blotting analysis confirmed that SID-PEMFPeps significantly regulated the expression of key proteins in TLR4/NF-κB/MAPK signaling pathways, which indicated that SID-PEMFPeps could delay D-Gal-induced senescence of PC12 cells (Figure 11).

In this study, the senescence-delaying effects of PEMFP, PEMFPeps, and SID-PEMFPeps were investigated by establishing a D-Gal-induced PC12 cell model for the first time. The activity of PEMFPeps was obviously increased after simulated digestion, which indicated that PEMFPeps and SID-PEMFPeps could provide a solid data foundation for the prevention of aging and neurodegenerative diseases. However, there are still more issues to explore in the future. For example, experiments should be conducted on the permeability of PEMFPeps and SID-PEMFPeps through epithelial cell experiments in order to validate the effects further. Moreover, during the growth of *Pleurotus eryngii*, the types of carbon and nitrogen sources, environment temperature and humidity, ventilation capacity, light, and pH value could all influence the nutritional and protein composition. Thus, the impacts of these factors could be further investigated. Different extraction conditions could influence the extraction rate of PEMFP, such as solid–liquid ratio, extraction time, extraction temperature, and NaCl solution concentration. Moreover, different enzymes could influence the cleavage location of the protein and produce different peptides, so enzymatic methods could also be further studied. And the corresponding experimental conditions, such as pH, enzyme concentration, enzymatic hydrolysis temperature, and enzymatic hydrolysis time, also have great impacts on the properties and bioactivities of peptides. Thus, single-factor and response surface experiments could be conducted to obtain the optimal conditions. Moreover, the mechanisms associated with aging were complex, and transcriptomic and metabolomic analyses could be applied to screen other possible signal pathways. And in vivo experiments could also be deeply investigated in order to further verify the activity of PEMFPeps.

## Figures and Tables

**Figure 1 foods-13-03668-f001:**
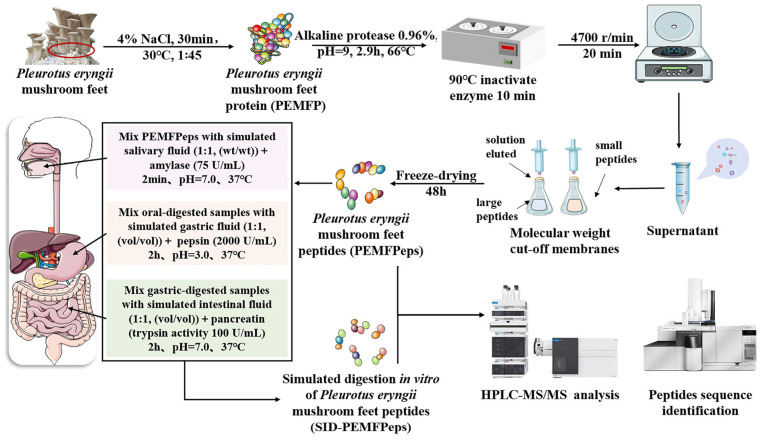
Preparation of PEMFP, PEMFPeps, and SID-PEMFPeps. Peptide sequence identification of PEMFPeps and SID-PEMFPeps.

**Figure 2 foods-13-03668-f002:**
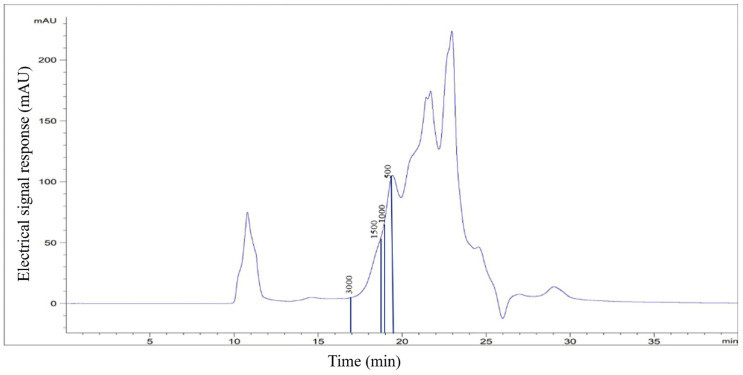
The molecular weight distribution of PEMFPeps. Standards are insulin (5733 Da), bacitracin (1422 Da), Gly-Gly-Tyr-Arg (451 Da), and Gly-Gly-Gly (189 Da), respectively.

**Figure 3 foods-13-03668-f003:**
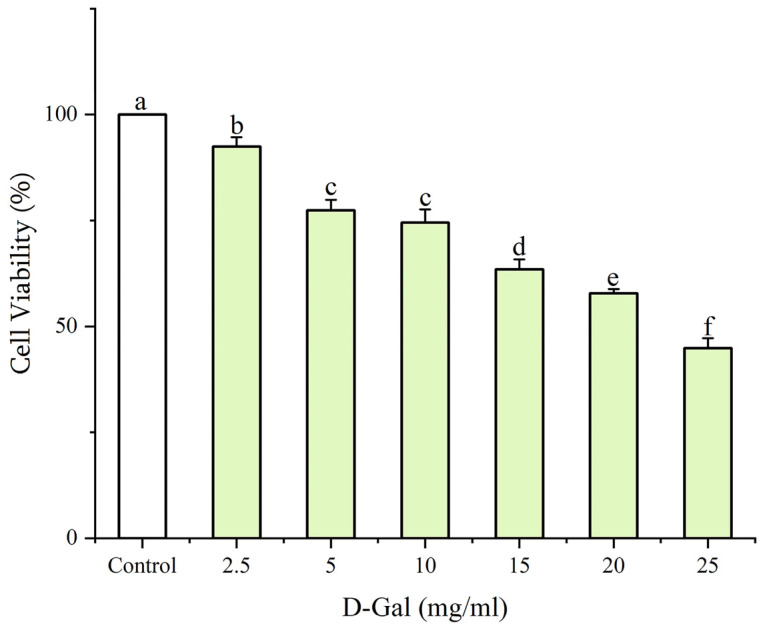
Cell viability of PC12 cells treated with D-Gal. Results were shown as means ± standard deviation (SD, *n* = 3). Bars with the same superscript letters are not significant (*p* > 0.05), and bars with different superscript letters are significant (*p* < 0.05).

**Figure 4 foods-13-03668-f004:**
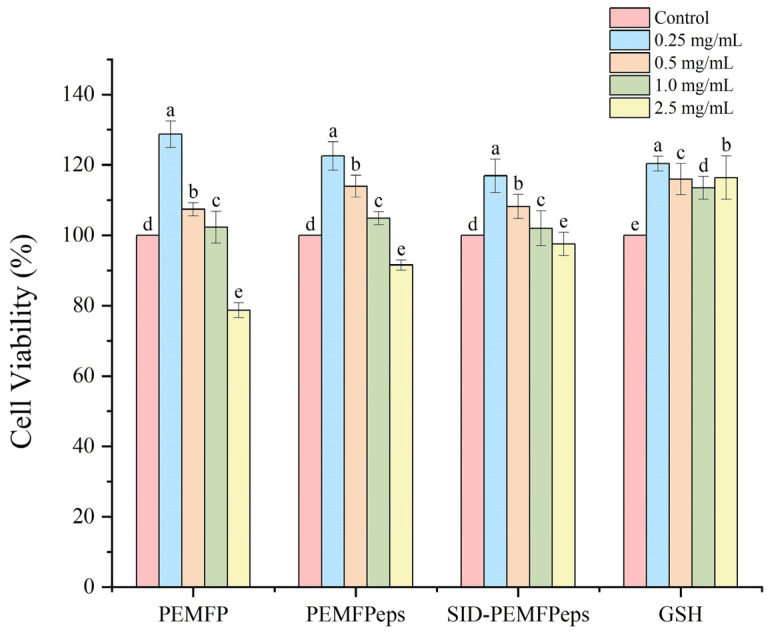
Cell viability of PC12 cells treated with PEMFP, PEMFPeps, SID-PEMFPeps, and GSH. Results were shown as means ± standard deviation (SD, *n* = 3). Bars with the same superscript letters are not significant (*p* > 0.05), and bars with different superscript letters are significant (*p* < 0.05).

**Figure 5 foods-13-03668-f005:**
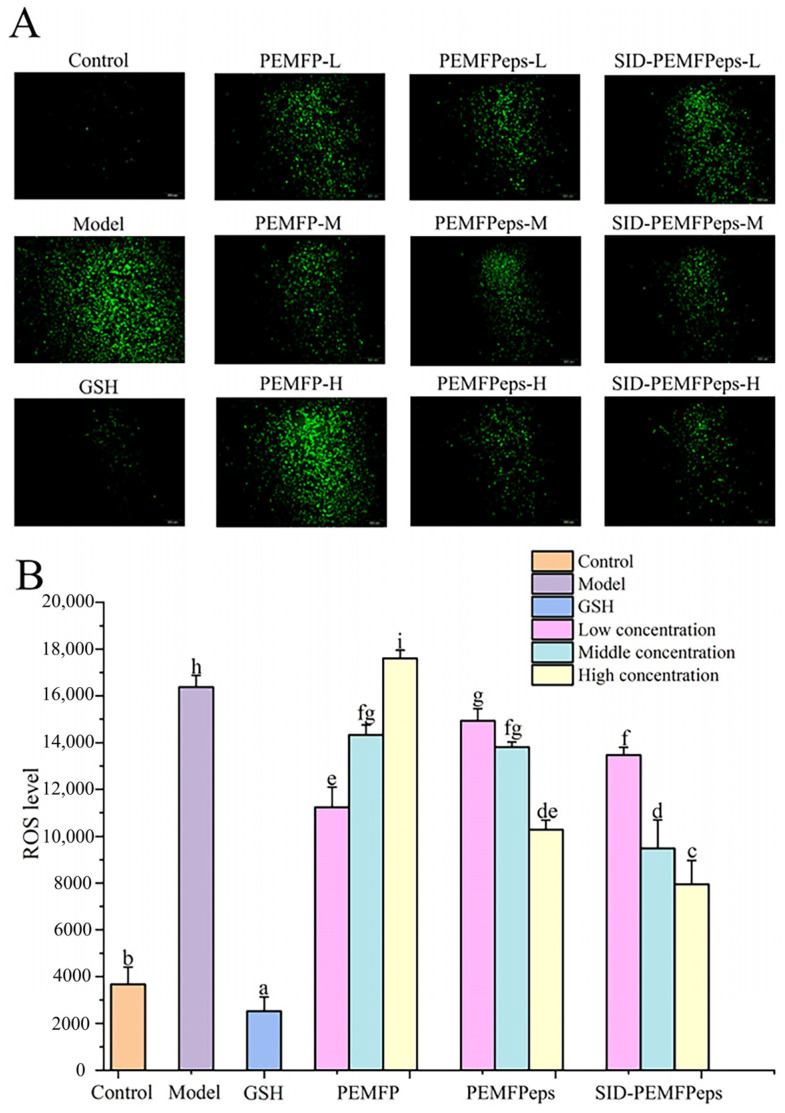
Fluorescence intensity (**A**) and ROS production (**B**) of D-Gal-induced senescent PC12 cells in the control group, model group, GSH group, and each sample group. Results were shown as means ± standard deviation (SD, *n* = 3). Bars with the same superscript letters are not significant (*p* > 0.05), and bars with different superscript letters are significant (*p* < 0.05).

**Figure 6 foods-13-03668-f006:**
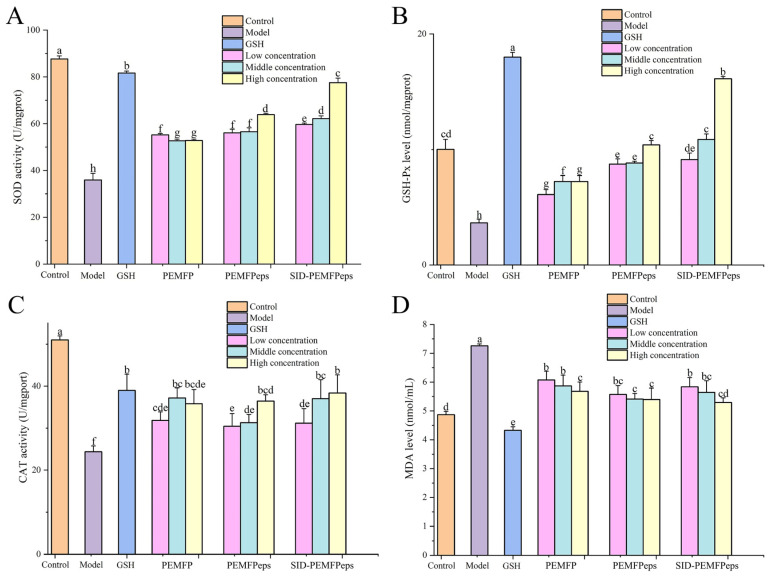
SOD (**A**), GSH-Px (**B**), CAT (**C**), and MDA (**D**) contents of PC12 cells in different groups. Results were shown as means ± standard deviation (SD, *n* = 3). Bars with the same superscript letters are not significant (*p* > 0.05), and bars with different superscript letters are significant (*p* < 0.05).

**Figure 7 foods-13-03668-f007:**
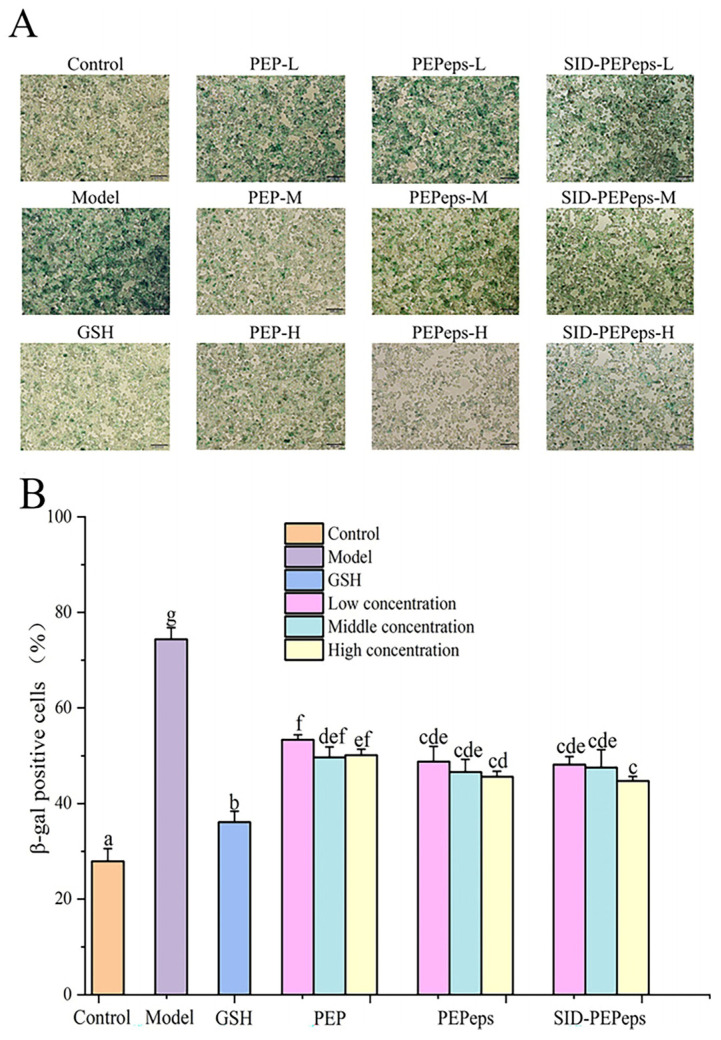
SA-β-gal staining (**A**) and the rate of SA-β-gal-positive cells (**B**) of D-Gal-induced PC12 cells in different groups. Results were shown as means ± standard deviation (SD, *n* = 3). Bars with the same superscript letters are not significant (*p* > 0.05), and bars with different superscript letters are significant (*p* < 0.05).

**Figure 8 foods-13-03668-f008:**
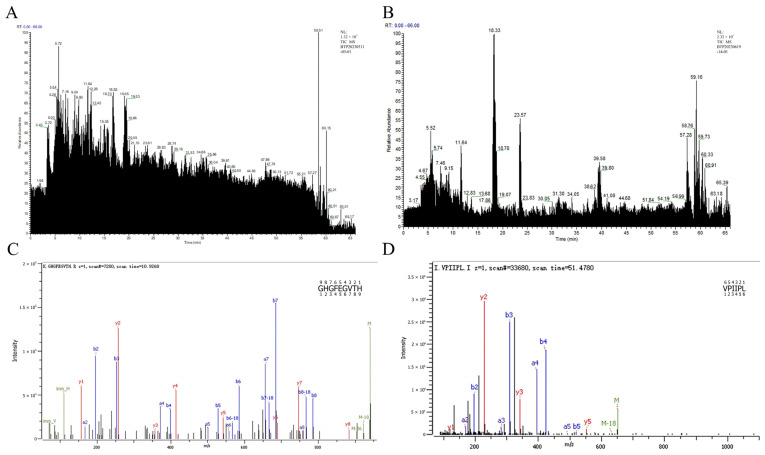
MS spectra of PEMFPeps (**A**) and SID-PEMFPeps (**B**). MS/MS spectra of new peptides of GHGFEGVTH (**C**) and VPIIPL (**D**).

**Figure 9 foods-13-03668-f009:**
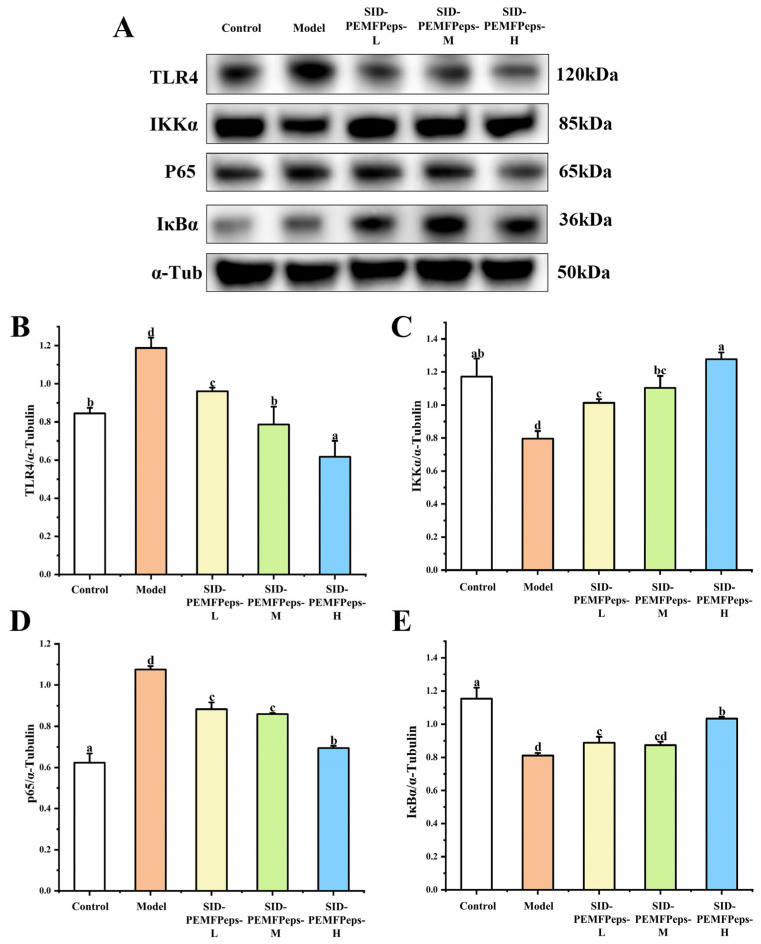
Western blotting analysis of key proteins in TLR4/NF-κB signaling pathways (**A**); data visualized from Western blot analysis of TLR4 (**B**), IKKα (**C**), p65 (**D**), and IκBα (**E**). Bars with the same superscript letters are not significant (*p* > 0.05), and bars with different superscript letters are significant (*p* < 0.05).

**Figure 10 foods-13-03668-f010:**
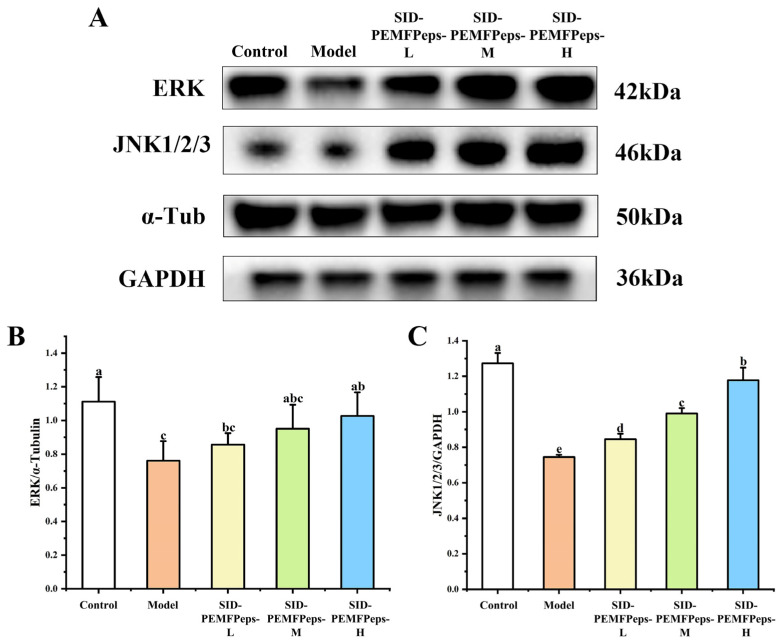
Western blot analysis of key proteins in the MAPK signaling pathway (**A**) and data visualized from Western blot analysis of ERK (**B**) and JNK1/2/3 (**C**). Bars with the same superscript letters are not significant (*p* > 0.05), and bars with different superscript letters are significant (*p* < 0.05).

**Figure 11 foods-13-03668-f011:**
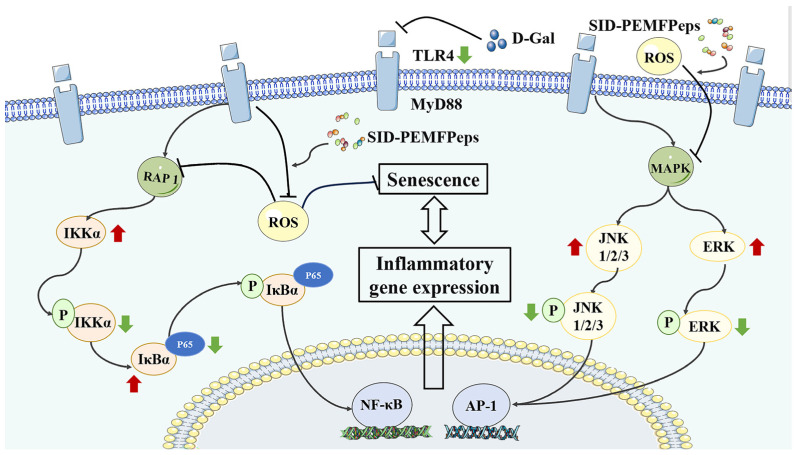
Mechanisms of SID-PEMFPeps on delaying D-Gal-induced senescence of PC12 cells through TLR4/MAPK/NF-κB signaling pathways. Red arrows indicate increased protein expression and green arrows indicate decreased protein expression.

**Table 1 foods-13-03668-t001:** Main peptide sequences of PEMFPeps.

Peptide Sequence	Molecular Weight (Da)	Intensity	Amino Acid Number	Hydrophobic Amino Acid Number	Aromatic Amino Acid Number	Polar Amino Acid Number	Total Number of Key Amino Acids	Score
GHGFEGVTH	940.425	1.37 × 10^7^	9	5	1	2	7	636.4
GRHTGPGKR	965.537	1.57 × 10^8^	9	4	0	4	8	601.4
DDESAIGIR	975.474	1.64 × 10^7^	9	4	0	1	5	569.6
HTGNIPLDE	995.477	1.14 × 10^7^	9	4	0	1	5	564.9
DACLPSPK	887.427	8.88 × 10^7^	8	4	0	1	5	564.2
GPPGTGKTL	827.460	2.90 × 10^7^	9	6	0	1	7	557.3
SDEQISLK	919.474	7.45 × 10^6^	8	2	0	1	3	555.8

**Table 2 foods-13-03668-t002:** Main peptide sequences of SID-PEMFPeps.

Peptide Sequence	Molecular Weight (Da)	Intensity	Amino Acid Number	Hydrophobic Amino Acid Number	Aromatic Amino Acid Number	Polar Amino Acid Number	Total Number of Key Amino Acids	Score
LLLH	495.328	3.27 × 10^7^	4	3	0	1	4	336.9
VPIIPL	651.442	2.60 × 10^7^	6	6	0	0	6	319.0
HVPL	465.282	1.29 × 10^8^	4	3	0	1	4	314.6
PIIPL	552.375	3.73 × 10^7^	5	5	0	0	5	266.1
LLLHI	608.413	6.95 × 10^6^	5	4	0	1	5	255.2
VHVPLS	651.381	2.53 × 10^8^	6	4	0	1	5	226.2
VPIIP	538.359	1.08 × 10^8^	5	5	0	0	5	206.0

## Data Availability

The original contributions presented in this study are included in the article. Further inquiries can be directed to the corresponding author.

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
