# Peer review of "Identification of Peptides from Edible Pleurotus eryngii Mushroom Feet and the Effect of Delaying D-Galactose-Induced Senescence of PC12 Cells Through TLR4/NF-κB/MAPK Signaling Pathways"

_foods, 2024, doi:10.3390/foods13223668_

Round 1

Reviewer 1 Report

Comments and Suggestions for Authors

Line 100: The abbreviation of hydrogen peroxide is written incorrectly.

Iine 125: Why boiling water was used for resuspension of PEMFP powder?

Line 127: What do you mean with 2.9 h? How this value is chosen?

Line 180: Please re-write the sentences with avoiding the repetition of numbers.

Line 248: Although molecular weight is an important parameter in antioxidant activity of peptides, it is not the only factor. Concluding that a protein hydrolysate has high antioxidant activity based solely on the presence of small peptides would be incorrect.

Figure 4: How you can explain the lower protective effects of PEMFP, PEMFPeps, and SIS-PEMFPeps at higher concentrations?

Figure 5: How you can explain the greater effectiveness of samples in reducing ROS level at lower concentrations?

Figure 5: Please indicate what are control and model in figure‘s caption.

Figure 6 D: It is shown that the level of GSH-PX in GSH treated cells is even higher than its value in control cell. How can you explain this observation?

Line 481: The sentence “consumption by oral administration may have better effect“is not clear. Better effect than what?

Line 489: The oral administration of PEMFPeps is suggested to prevent the progression of aging based on the results of comparing the activity of PEMFPeps and SID-PEMFPeps. How this conclusion can be made with no information about the permeability of produced peptides through epithelial cells?

The Mass spectra of all identified peptides should be included as supplementary file.

Reviewer 2 Report

Comments and Suggestions for Authors

Peptides derived from food sources may exert beneficial biological activity on processes related to cellular senescence and aging, and studies in this area are relevant. However, some issues should be considered in this manuscript, such as:
1) The objective of the study should be described more clearly in the “Abstract”;
2) The paragraphs of the introductory text could be better organized according to the key points of the investigation, such as cellular senescence and the aging process, the nutritional and functional value of edible mushroom species considering mainly their protein content, and an overview of the literature on the properties and bioactivity of peptides derived from Pleurotus eryngii mushrooms, especially related to their anti-aging potential;
3) Regarding the methodology, add more detailed information on the definition/selection of the peptide concentrations used in the study and on the experimental groups (controls and treatment);
4) The results obtained should be widely discussed by the Authors considering data available in the current literature. I suggest that the Authors expand the discussion and provide a broad approach on the factors that influence the nutritional and protein composition of edible mushrooms such as Pleurotus eryngii, the impact of these factors on the properties and bioactivity of proteins/peptides, experimental conditions and peptide extraction methods, and mechanisms associated with aging.

Reviewer 3 Report

Comments and Suggestions for Authors

Identification of new peptides from edible Pleurotus eryngii mushroom feet and the effect on delaying D-Galactose induced senescence of PC12 cells through TLR4/NF-κB/MAPK signaling pathways

Title

Please explain why are new peptides?, in my opinion this word is not correct, to be better this word should be eliminated               

Introduction

Please, the authors could explain why is important inhibit the senescense-associated

-b-galactosidase (SA-b-gal) activity?

Lines 53-54. Please, Could the authors check that paragraphs because it is not clear

Material and Methods

In general in this part of the document is a lack of references

Suggestions

Line 17 script for in vitro of Pleurotus eryngii mushroom

Line 25 defines ROS

Line 34 change to script all the in vitro term

Line 61 AChE activity to what is referred to as this activity

Line 24 Describe the enzymes in the abstract because they were mentioned on line 98

Line 100 could it be better catalase instead of Hydrogen peroxide?

Increase the size of the Figure 5 A and Figure 7 A cells

In the first column of Table 1 could the word sequence go together?

Lines 247-248.  Please explain why molecular weights less than 1000 Da indicate good antioxidant activity.

327-339. Please, explain why the mushroom peptides have this effect.

Round 2

Reviewer 1 Report

Comments and Suggestions for Authors

All comments and remarks have been addressed in the manuscript and it is suitable for publication in current format.

Author Response

Thank you very much for your comments and support.

Reviewer 2 Report

Comments and Suggestions for Authors

The aspects mentioned above in item 4 could be better addressed by the Authors considering the current literature. I suggest that the discussion be expanded. Furthermore, the adjustments improved the manuscript.

Round 3

Reviewer 2 Report

Comments and Suggestions for Authors

The adjustments were sufficient.